# Bivariate sea-ice assimilation for global ocean Analysis/Reanalysis

Andrea Cipollone[1], Deep Sankar Banerjee[1,2], Doroteaciro Iovino[1], Ali Aydogdu[1], and Simona Masina[1]

[1]Ocean Modeling and Data Assimilation Division, Centro Euro-Mediterraneo Sui Cambiamenti Climatici, Bologna, Italy
[2]presently at: Plymouth Marine Laboratory, Plymouth, UK and National Centre for Earth Observation, Plymouth, UK

**Correspondence:** Andrea Cipollone (andrea.cipollone@cmcc.it)

**Abstract.** In the last decade, various satellite missions have been monitoring the status of the Cryoshopere and its evolution. Beside sea-ice concentration data, available since the 80s, sea-ice thickness retrievals are now ready to be used in global operational prediction and global reanalysis systems. Nevertheless, while univariate algorithms are commonly used to constrain sea-ice area or volume, multivariate approaches are yet to be employed due to highly non-Gaussian distribution of sea-ice variables together with the low accuracy of thickness observations. This study extends a 3Dvar system, called OceanVar and routinely employed in the production of global/regional operational/reanalysis products, to process sea-ice variables. The tangent/adjoint versions of an anamorphosis operator, are used to transform locally the sea-ice anomalies into Gaussian control variables and back, minimising in the latter space. The benefit brought by such transformation is described. Several sensitivity experiments are carried out using a suite of diverse datasets. The sole assimilation of the CryoSat-2 provides a good spatial representation of thickness distribution but still overestimates the total volume that requires the inclusion of SMOS data to converge towards the observation estimates. The intermittent availability of thickness data can lead to potential jumps in the evolution of the volume and requires a dedicated tuning. The use of the merged L4 product CS2SMOS shows the best skill score when validated against independent measurements during the melting season when satellite data are not available. This new sea-ice module is meant to simplify he future coupling with ocean variables.

## 1 Introduction

The recent availability of sea-ice thickness retrievals have offered a unique opportunity to significantly improve the reconstruction of the past state at high latitudes as well as its prediction. Thickness estimates were first derived from the ERS-1/ERS-2 radar altimetry echoes between 1993 and 2001 in a pioneering reconstruction of Arctic sea-ice thickness distribution up to 81.5°N (Laxon et al., 2003). In 2003 a dedicated satellite mission ICESat was launched to monitor the thinning of Arctic ice (Forsberg and Skourup, 2005). More recent missions consider the Soil Moisture and Ocean Salinity (SMOS) mission in 2009 (Kaleschke et al., 2010; Tian-Kunze et al., 2014), the polar-orbiting CryoSat-2 in 2010 (Wingham et al., 2006) and the ICESat-2 mission in 2018 (Kwok et al., 2019; Petty et al., 2022). Most of these datasets are yet to be harnessed by present reanalysis systems as pointed out by recent reanalysis inter-comparison studies that show large discrepancies in several sea-ice features despite a rather general agreement in the sea-ice extent (Chevallier et al., 2017; Uotila et al., 2019; Iovino et al., 2022). Thickness data could be also employed to ameliorate short and long-term prediction: the memory of a realistic thickness distribution within the initial conditions has been shown to persist well beyond the seasonal timescale (Day et al., 2014;

Blanchard-Wrigglesworth et al., 2017). Despite that, the intermittent occurrence of such data during the year, the large errors associated to them (Zygmuntowska et al., 2014) and the highly non-Gaussian distributions of sea-ice related uncertainties, made the multivariate assimilation of sea-ice data still an active research field. Nowadays, the sole assimilation of the sea-ice concentration in a univariate fashion, is a well-established approach (Posey et al., 2015; Lemieux et al., 2016; Zuo et al., 2019). Preliminary studies on the addition of a second univariate assimilation scheme for thickness have come out only recently at global level. Blockley and Peterson (2018); Mignac et al. (2022) showed the benefit of using of CryoSat-2 and later CryoSat-2/SMOS data to correct the Arctic thickness distribution, exploiting a variational approach within the FOAM system. They also point out the need of a better estimation of SIT observation errors. At regional scale, multivariate approaches were developed, Xie et al. (2016, 2018) confirm the benefits of the assimilation of CryoSat-2 and SMOS in the TOPAZ regional forecast system based on the Ensemble Kalman filter. The main correction comes from the use of CryoSat-2 data, the assimilation of SMOS reduced the error in the thin-ice of about 11 and 22% in March and in November respectively, without degradation in the other variables. Yang et al. (2014); Mu et al. (2018b) tested the Localized Singular Evolutive Interpolated Kalman filter to integrate thickness data and showed an overall error that is similar to the PIOMAS system (Zhang and Rothrock, 2003) when compared to independent in-situ measurements.Finally, Cheng et al. (2023) has recently showed in a standalone Lagrangian sea-ice model, neXtSIM, interfaced to a deterministic EnKF scheme in a multivariate manner that improvements in SIT estimates indicate the importance of assimilating weekly CS2SMOS SIT while the improvements of SIC and ice extent are moderate but benefit from daily correction from OSI-SAF SIC. In this study, we extend an operational 3DVar data assimilation (DA) scheme, OceanVar, employed in the production of global and regional ocean reanalysis and forecasts (Storto et al., 2019a; Escudier et al., 2021; Lima et al., 2021; Ciliberti et al., 2022), to treat sea-ice concentration (SIC) and thickness (SIT) data. The novelty in this approach relies on the inclusion of tangent/adjoint version of an anamorphosis operator in the control vector transformation to deal with the breaking of the Gaussian assumption of sea-ice variables (Brankart et al., 2012; Simon and Bertino, 2009; Béal et al., 2010). The operator transforms the probability density functions of SIC/SIT anomalies towards Gaussian-like ones performing the minimization in this space. It is originally based on the tool made available by the SANGOMA project (http://www.data-assimilation.net/) further adapted for the bivariate assimilation of SIC/SIT within the OceanVar framework. While being able to maintain the correct cross-correlation between the two parameters, such operator is also able to preserve the strong spatial anisotropy of sea-ice variables close to the edge. Several sensitivity experiments were carried out with the new scheme assimilating different thickness products: SMOS, CryoSat-2 and optimally-interpolated product CS2SMOS (Ricker et al., 2017), jointly with SIC data. Strategies to avoid discontinuities at the onset of the accretion period when the SIT data starts to be available are discussed. The paper is organized as follows: Section 2 provides a description of the observation-based datasets used in this study and the ocean/sea-ice models employed. In Section 3 we detail the new module of OceanVar that deals with sea-ice variables. The comparison among different DA set up and observations are discussed in Section 4 by means of a suite of ad-hoc metrics together with the independent validation of thickness field against mooring and airborne data. The relative influence of the observation networks is also assessed. Conclusions and remarks are drawn in Section 5.

## 2 Data and Models

In the past few decades, several satellite-derived datasets of Arctic sea ice thickness have been disseminated mainly limited to the freezing season (October-April in the Arctic) due to the difficulty in discerning signals from open water and meltponds during the melting season. Radar altimeters installed on the polar-orbiting CryoSat-2 (Laxon et al., 2013; Hendricks and Ricker, 2020) provide thick sea-ice data, typically thicker than 0.5m (Zygmuntowska et al., 2014), by relying on the knowledge of the snow depth (Warren et al., 1999) and on the assumption of hydrostatic equilibrium (Ricker et al., 2014; Tilling et al., 2016). Measurements of thin sea ice, roughly up to 0.5m, are instead extracted from passive microwave radiometer (Huntemann et al., 2014), within the European Space Agency (ESA) Soil Moisture and Ocean Salinity (SMOS) mission, analysing the satellite brightness temperature in the L-Band microwave frequency (Kaleschke et al., 2010). The complementarity characteristics of these two products fostered the generation of a weekly optimally interpolated merged product called CS2SMOS http://data.meereisportal.de (Grosfeld et al., 2016; Ricker et al., 2017) that is released together with a mapping error accounting for merging and interpolation processes. As shown by Xie et al. (2018) such error can be used as a first guess to construct a better observation error following Desroziers' method (Desroziers et al., 2005). In the context of observation-derived datasets, is worth to mention the recent availability of a year round product that extimates summer-time thickness using deep learning methods (Landy et al., 2022) that however will be not considered in the present analysis. Jointly with SIT data, daily concentration measurements, computed from SSMIS (2006-2015) instruments with atmospheric corrections from ERA-Interim (Lavergne et al., 2019) and reprocessed by Ocean and Sea Ice Satellite Application Facility (OSISAF, 2021), are assimilated. The ocean/sea-ice configuration follows the global set-up employed in the C-GLORS reanalysis production (Storto and Masina, 2016). The ocean model is NEMO v3.6 (Madec, 2016) coupled with the Louvain-la-Neuve sea-ice model LIM version 2 (Fichefet and Maqueda, 1997), a three-layer (two layers of sea ice and one of snow) thermodynamic-dynamic model which here employs the elasto-visco-plastic rheology (Bouillon et al., 2009) and one thickness category. The use of a multi-category sea-ice model is foreseen in the next future, providing a more complex representation of the sea-ice interaction with the other components of the earth system. The Ice Thickness Distribution, ITD, (Thorndike et al., 1975) accounts for the sub-grid (unresolved) physics in a statistical sense: internal/external thermodynamic/mechanic processes can change the total thickness as well as its distribution, the latter being only partially parametrized by simpler mono-category sea-ice model. On the other hand, the practical discretization of such categories as well as their number should be properly tuned to contain the computational cost and still provide benefits with respect the mono-category models. In (Uotila et al., 2017) the Authors compare a set of simulations performed with multi- and mono-category sea-ice models: LIM3 and LIM2 respectively. They showed that the decline of Arctic sea-ice extent in the last decade as well as Antarctic seasonal variability are better reproduced with LIM3. However, the impact on the ocean sector is usually very small. Moreover, the discretization has a significant impact on the mean state (Massonnet et al., 2019) and it can be inferred that the optimal configuration is different for Arctic and Antarctic sea-ice. In this context the coupling with a sea-ice DA system could help in reducing the differences between multi/mono category models. A tuned multi-category model can ease the effort of DA and provide a consistent realistic representation of such variables not directly corrected by the DA. The present configuration uses a tripolar grid with nominal horizontal resolution of 1/4°, i.e. 25

95  km at the equator increasing toward the poles with 75 vertical levels and partial steps at the bottom (Barnier et al., 2006). The sea-ice and ocean model are forced by hourly ERA5 atmospheric reanalysis (Hersbach et al., 2020) with horizontal resolution of $0.25°$ using 10-m wind, 2-m temperature and humidity, short and long radiative fluxes, precipitation and snow. The coupling frequency between the sea-ice and ocean model is one hour.

## 3   Data assimilation scheme

Variational schemes can be described in a purely statistical sense, following a Bayesian formulation, where the model variability is interpreted as a stochastic error that follows a spatial- and time-varying probability density function (pdf) as in Carrassi et al. (2018). The best ocean state is defined as the mode of the a-posteriori pdf of the ocean state conditioned to the presence of observations. Under the hypothesis of normal distribution, this translates in seeking the minimum of the following cost function,

$$J(\delta\boldsymbol{x}) = \frac{1}{2}\delta\boldsymbol{x}^{\mathrm{T}}\boldsymbol{B}^{-1}\delta\boldsymbol{x} + \frac{1}{2}(\boldsymbol{H}\delta\boldsymbol{x} - \boldsymbol{d})^{\mathrm{T}}\boldsymbol{R}^{-1}(\boldsymbol{H}\delta\boldsymbol{x} - \boldsymbol{d}) \tag{1}$$

where the first addend comes from the pdf of the anomalies with respect to the initial background state, while the second refers to the pdf of the observations conditioned to the model background. Eq. (1) is the standard incremental formulation of the cost function found in the OceanVar scheme (Dobricic and Pinardi, 2008) where $\delta\boldsymbol{x}$ labels the increments that corresponds to the difference between the final analysis state $\boldsymbol{x}_a$ and the initial ocean state $\boldsymbol{x}_b$ in the minimum of the cost function. $\boldsymbol{B}$ and $\boldsymbol{R}$ are the
background- and observation-error covariance matrices respectively, $\boldsymbol{d}$ is the vector of misfits calculated using the non-linear observation operator, $\boldsymbol{H}$ is the tangent-linear version of the observation operator. The inclusion of sea-ice variables implies the augmentation of the ocean state vector, initially composed by $x \sim (T, S, SLA)$ with the addition of sea-ice concentration and thickness $x \sim (T, S, SLA, SIC, SIT)$. Being the minimisation problem unconstrained, a pre-conditioning is applied by a Control Vector Transformation or CVT ($\boldsymbol{V}$) that moves the minimization in a control space $\boldsymbol{v}$ defined as $\boldsymbol{\delta x} = \boldsymbol{V}\boldsymbol{v}$ and
$\boldsymbol{B} = \boldsymbol{V}\boldsymbol{V}^{\mathrm{T}}$. Such $\boldsymbol{B}$ matrix is at the basis of any filtering process, it spreads information in areas where no or sparse data are present and smooths information in observation-dense regions. In literature different methodologies are used to shape sea-ice background error covariances $\boldsymbol{B}$, from multivariate ensemble-based methods (Xie et al., 2016) to univariate approaches based on historical simulations (Zuo et al., 2019) or to short hindcast runs (Fiedler et al., 2021; Mignac et al., 2022). Following the construction of $\boldsymbol{B}$ for ocean variables in OceanVar (Dobricic and Pinardi, 2008; Storto et al., 2010), such control vector
transformation $\boldsymbol{V}$ is composed by a sequence of linear operators coming from both physical balances and statistical methods to add complexity in the covariance matrix $\boldsymbol{B}$:

$$\boldsymbol{V} = \boldsymbol{V}_{\mathrm{gICE}\to\mathrm{ICE}}\boldsymbol{V}_{\eta}\boldsymbol{V}_{h}\boldsymbol{V}_{V(T:S:\mathrm{gSIC}:\mathrm{gSIT})} \tag{2}$$

where $\boldsymbol{V}_{\eta}$ is the dynamic height balance converting increments of temperature and salinity into increments of sea level through local hydrostatic balance (Storto et al., 2010), $\boldsymbol{V}_{h}$ models the horizontal correlations through the application of a recursive
filter, $\boldsymbol{V}_{V}$ is the vertical covariance operator made by empirical orthogonal functions (EOFs) and $\boldsymbol{V}_{\mathrm{gICE}\to\mathrm{ICE}}$ is the linearised

anamorphosis operator that transforms the Gaussian sea-ice variables into physical ones, gICE refers to both the operators applied independently to gSIC and gSIT. Sea-ice variables are not directly covaried with the other variables, the break of Gaussianity in fact can generate unrealistic corrections in a multivariate framework due to the poor linear relationship driven by a simple covariance matrix (Bertino et al., 2003; Brankart et al., 2012). A similar approach has been previously employed in literature to deal with strongly non-Gaussian variables (Simon and Bertino, 2009; Béal et al., 2010) and is presented here for dealing with SIC and SIT fields. The $V_{\text{gICE}\rightarrow\text{ICE}}, V^{\text{T}}_{\text{ICE}\rightarrow\text{gICE}}$ operators are the tangent and adjoint version of an anamorphosis operator developed and made freely available through the SANGOMA project that constructs such transformation empirically by mapping the different quantiles of the initial and final distributions (Brankart et al., 2012).

Neglecting the ocean variables, the CVT transformation reduces to:

$$\delta\boldsymbol{x} = (\delta\text{SIC}, \delta\text{SIT}) = \boldsymbol{V}_{\text{gICE}\rightarrow\text{ICE}}\boldsymbol{V}_h\boldsymbol{V}_{(\text{gSIC:gSIT})}\boldsymbol{v} \tag{3}$$

Firstly the gSIC/gSIT are cross-correlated through $\boldsymbol{V}_{(\text{gSIC:gSIT})}$, then increments are spread horizontally through the recursive filter operator $\boldsymbol{V}_h$. The final fields are transformed into physical variables through $\boldsymbol{V}_{\text{gICE}\rightarrow\text{ICE}}$.

### 3.1 Background error covariance matrix

The benefits brought by the anamorphosis transformation have been already discussed in literature: linear correlations in the transformed space can be seen as a non parametric correlation in the original space, being more adequate to treat nonlinear dependencies and more robust to the presence of outliers in the observations (Chilès and Delfiner, 1999; Corder and Fore-man, 2009; Brankart et al., 2012). The operator $\boldsymbol{V}_{\text{gICE}\rightarrow\text{ICE}}$ in Eq. (3) is spatially and monthly varying, computed at each model grid point by employing monthly fields of a historical 31 year-long NEMO-LIM2 simulation. Such operator requires the anamorphosis transformation to be locally continuous, in the case the numerical derivative does not exist the correspond-ing increment is zero and no correction is generated. To avoid the presence of discontinuous probability densities, that reflect an underestimation of the model error, we enrich the sample size for each point $(x, y)$ with values from neighbouring points $(x-1 : x+1, y-1 : y+1)$, therefore each initial distribution is shaped by $31 * 9 = 279$ samplings and then mapped to a normal distribution using a quantile mapping with 21 quantiles (Brankart et al., 2012). An example of the application of Gaussian anamorphosis on the SIT field is shown in Figures 1.a-d that display the initial and final map for two years 1999 and 2014. Similar procedures apply to SIC field (not shown). Gaussian variables can be interpreted as a "measure" of the anomalous content of the original variable given its pdf. Such anomaly is then normalised to a common scale, amplifying/reducing the variability in each point according to the imposed normal distribution. Panels a) and c) show the SIT and gSIT spatial dis-tributions for March 1999, respectively. The strong positive gSIT anomaly in the Siberian sector for March 1999 reflects the excess of sea-ice compared to the climatological March distribution. An opposite behaviour is seen in March 2014 (Figures 1.b,d) where gSIT values are more homogeneous and slightly negative, meaning that original spatial distribution is close to the climatological one, despite being uniformly lower in magnitude.

The cross-correlation (between SIC and SIT) is only slightly modified by this transformation as it can be inferred from Figures 1.e-h that compare the two fields prior and after the transformation for March and September. the spatial structure is

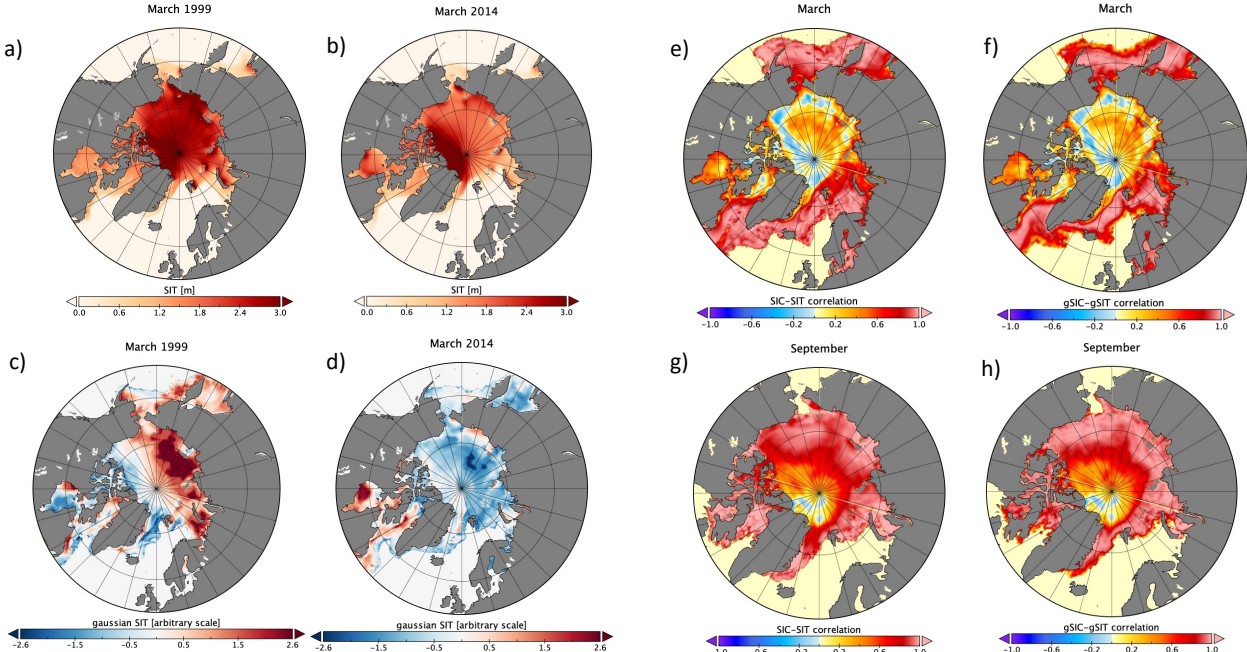

**Figure 1.** Panels a and b show the SIT spatial distribution for March 1999 and 2014 respectively, taken from the historical 31 year-long simulation used to construct the anamorphosis transformation. Panels c and d correspond to the SIT in gaussian space for the same dates. Panels e and f are the cross-correllation between SIC and SIT in physical and gaussian space respectively for March in each grid point. Panels g and h show the same as e,f for September

similar in the two cases, while the magnitude slightly differs especially in perimetrical areas where ice is seldom present and
the statistics less reliable. Two dynamically different regions emerge from these maps: i) a first zone with a high positive cross-correlation where an increase in concentration automatically generates a corresponding increase in thickness and viceversa; ii) a second zone where these two variables tend to disentangle and correlation drastically drops to zero. This last behaviour is typical of areas where the concentration is already close to 1 and the variation in thickness does not affect directly the concentration.

The use of local Gaussian space in each point of the grid turns to be crucial for a correct application of the horizontal correlation operator especially close to sea-ice edge. Figure 2 shows the sea-ice increments in a test case, says the third week of February 2015, generated with and without the application of $V_{gICE \to ICE}$ with a large fixed correlation length of 150km and three iterations of a first order recursive filter. Green solid line corresponds to the mean sea-ice edge in that week, SIC and SIT are jointly assimilated close to the sea-ice edge. In the physical space an isotropic spread of information towards
the ice-free areas is seen (Figures 2.c,d). The use of $V_{gICE \to ICE}$ "re-center" the increments (in the Gaussian space) within the range of physical values, reducing the wrong isotropic diffusion (Figures 2.a,b) and following the variability of the specific region. This operator seems to be crucial in the assimilation of sparse data and long horizontal correlation lengths. On the other

hand, the diffusion in physical space can provide good results in data-dense regions where the correlation length can be safely reduced to a small value. In the following we set a fixed value of 50 km that it has been shown to provide satisfactory results in a variational scheme (Mignac et al., 2022). The benefits brought by spatially and seasonally varying correlation lengths may be investigated in future. It is worth to note that the use of tangent/adjoint approximations of the anamorphosis leads the assimilation of extreme events, to be suboptimal (i.e. observations that are far from the background value). Tangent/adjoint approximations of any operator are valid in the proximity of the background value and become less and less accurate in the case of large corrections and highly non-linear operator. This is anyway a limitation that is implicit in any three-dimensional variational scheme. Moreover, the anamorphosis should span all the possible physical values in each grid point. In the case the background is out of the range of values used for the anamorphosis, then it is not possible to calculate the derivative and the corresponding increments are zero. This means that extreme events in the background (not present in the 31 years of simulation) do not receive corrections. In (Simon and Bertino, 2012) they include an exponential tail to the anamorphosis in order to treat values out of bounds. A further approach could be the use of a hybrid B where the ensemble part goes to update the anamorphosis with the inclusion of new model values as well as adding the "error-of-the-day".

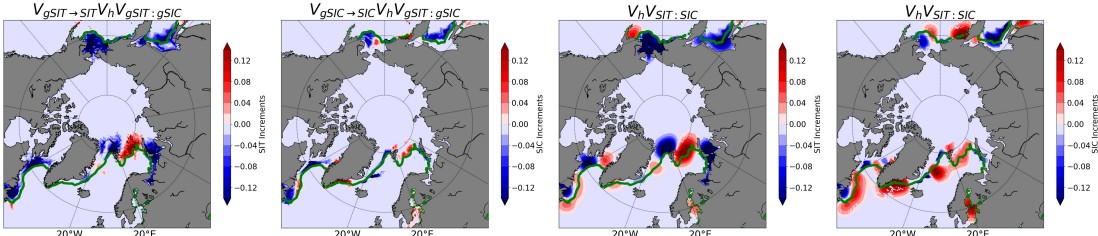

**Figure 2.** Examples of increments obtained from the joint assimilation of SIC and SIT data in different set up and close to sea-ice edge. First and second panels corresponds to SIT and SIC increments achieved by applying the anamorphosis transformation in a test case. Third and fourth panels refer to the same increments but without the transformation for the same date.

### 3.2 Observation error

The observation error (OE) includes different sets of uncertainties: instrumental errors, inaccuracies of the observation operators, unresolved dynamics, etc. (Oke and Sakov, 2008). Under the assumption of error independency the structure of $\boldsymbol{R}$ simplifies into a diagonal matrix that seems however sub-optimal in the case of dense datasets. A way to determine the presence of non-zero off-diagonal terms follows the implementation of Desroziers' relations (Desroziers et al., 2005) that combine model departures and assimilation residuals to diagnose the "correctness" of OE in observation space. Specifically the relation

$$\boldsymbol{E}[d_a^o(d_g^o)^T] = \boldsymbol{R} \tag{4}$$

links each element of the $\boldsymbol{R}$ matrix to a-posteriori statistical diagnostics, where $d_a^o$ being the residuals (analysis minus observations) while $d_g^o$ refers to the initial misfits (background minus observations). Desroziers' relations are generally used to

optimised the first guess OE (Xie et al., 2018) but can be also employed to add time-dependent effects both in $\boldsymbol{B}$ and $\boldsymbol{R}$

matrices (Storto and Masina, 2016; Escudier et al., 2021). It is worth to note that they must be used with caution because of

the presence of sampling errors and biases that can spoil the diagnostics (Ménard, 2016). Figure 3 shows these off-diagonal

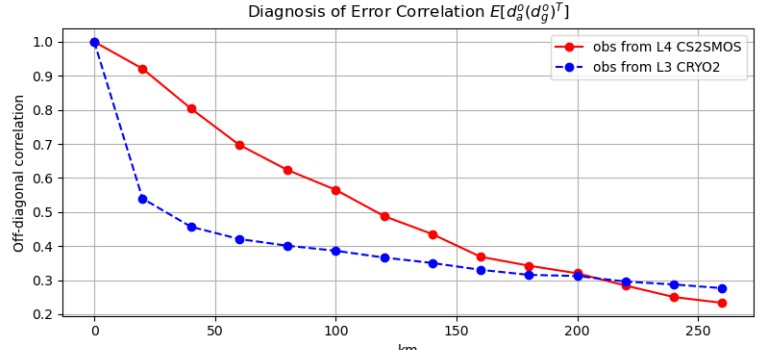

**Figure 3.** Correlation between different OE as function of the observation distance with a bin of 20km. Green line corresponds to L3CR2 experiment that assimilates only L3 Cryosat-2 data while red line shows the same for L4DE1 experiment (assimilation of L4 CS2SMOS data).

terms as function of the distance between observations and evaluated through the Equation (4). The green line refers to the

L3 CryoSat-2 data (experiment L3CR2 in table 1, see next section), while the red line labels L4 CS2SMOS data (experiment

L4DE). Statistics are averaged over a four-year-long reanalysis timeseries, after the application of the thinning procedure, and

restricted to 5000 observations per week (being the assimilation weekly). A minimum threshold of 0.1 m in thickness is im-

posed to avoid ice-free areas. The red line (CS2SMOS data) shows an error correlation that reduces slowly with the distance,

while a sudden drop is present for L3 CryoSat-2 data, demonstrating a much less interdependency among close errors. Several

studies are recently focused on different methods to include the error correlation in DA schemes (Storto et al., 2019b; Ruggiero

et al., 2016). At present, many operational systems further increase the Desroziers' OE to partially alleviate the absence of such

off-diagonal terms in $\boldsymbol{R}$ (Benkiran et al., 2021). However such solution requires some extra care for satellite data that are not

continuous over time as shown in the next section.

## 4   Results

Different data assimilation strategies are hereby discussed and compared. Table 1 summarises the main characteristics of each

experiment, while the DA set up differ, the model configuration remains identical. Ocean and sea-ice initial conditions refer to

the $1^{st}$ January 2011 from C-GLORS reanalysis (Storto and Masina, 2016).

**Table 1.** List of the four-year-long experiments performed with different data assimilation set up and observation employed.

| Exp Name | SIC data | SIT data | subsampling range | Desroziers' OE (multiplication factor ) |
|---|---|---|---|---|
| CTRL | None | None | None | None |
| L4DE1 | OSISAF | L4 CS2SMOS | None | 1 |
| L4DE30 | OSISAF | L4 CS2SMOS | None | 30 |
| L4SUB | OSISAF | L4 CS2SMOS | SIT $\sim$100km | 1 |
| L3CR2 | OSISAF | L3 CryoSat-2 | None | 2 |
| L3CR2&SM | OSISAF | L3 CryoSat-2 & SMOS | None | 2;2 |
| SICDE1 | OSISAF | None | None | 1 |
| SICDE02 | OSISAF | None | None | 0.2 |

## 4.1 Concentration data and sea-ice extent

The seamless presence of SIC data over the years, covering the full meteorological era, does not strictly require any ad-hoc optimisation to avoid discontinuities in the total sea-ice area. Figure 4 shows the evolution of the sea-ice area along the four-year
run for different set up and compared to OSISAF data. The free run, namely CTRL, has an overall Root-Mean-Square-Error (RMSE) of about $1.1 \times 10^6 \text{km}^2$ and $2.0 \times 10^6 \text{km}^2$ for the Arctic and Antarctic regions respectively. The use of SIC data decreases such error down to about $0.40 \times 10^6 \text{km}^2$ and $1.3 \times 10^6 \text{km}^2$, improving also the representation of trends during the growing and melting seasons. The two experiments SICDE1 and SICDE02 (OE is reduced to $1/5^{th}$) shows similar skill scores.

To compare the position of the sea-ice edge in the different experiments, the Integrated Ice Edge Error metric (IIEE) is
generally used (Goessling et al., 2016). The IIEE sums up all grid cell areas where models and observations are in disagreement on the presence or absence of sea-ice, with a concentration threshold of 15% (Blockley and Peterson, 2018). The assimilation of SIC data considerably decreases the sea-ice edge error compared to free run, with IIEE of about $1 \times 10^6 \text{ km}^2$ and $1.7 \times 10^6$ $\text{km}^2$ for Arctic and Antarctic regions respectively, while CTRL being around 1.6 and 2.6 $\times 10^6 \text{ km}^2$ (Figure 5). More than the 65% of the CTRL IIEE comes from an excess of sea-ice in ice-free areas (not shown). A noticeable improvement is seen in
August 2012 with CTRL peaking at 3.5 $\times 10^6 \text{ km}^2$ (with an overestimation of 2.5 $\times 10^6 \text{ km}^2$) that is reduced to 1.4 $\times 10^6 \text{ km}^2$ (overestimation of $0.8 \times 10^6 \text{ km}^2$) in the DA experiments. No significant differences are seen between SICDE1 and SICDE02 for concentration-related quantities. The frequency of the assimilation (weekly) does not seem able to remove concentration in regions where the model advects ice or where freezing conditions are met. A joint correction of ice and ocean variables in a multivariate approach can probably improve the skill by changing the sea surface temperature and salinity field as well.
Considering the impact of SIC assimilation on SIT field, the smaller OE in SICDE02 experiment leads to a larger correction in the thickness field thus spoiling the spatial distribution (see Section 4.2).

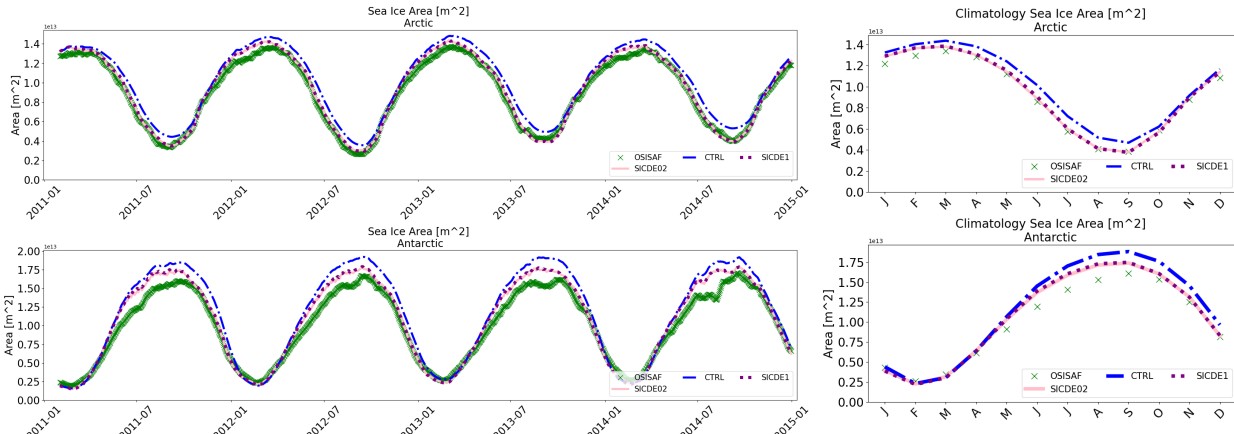

**Figure 4.** First and second rows show the timeseries of sea-ice area for different experiments (Table 1) compared to data from OSISAF in Arctic and Antarctic regions respectively. The corresponding seasonal variability is shown in the panels on the right.

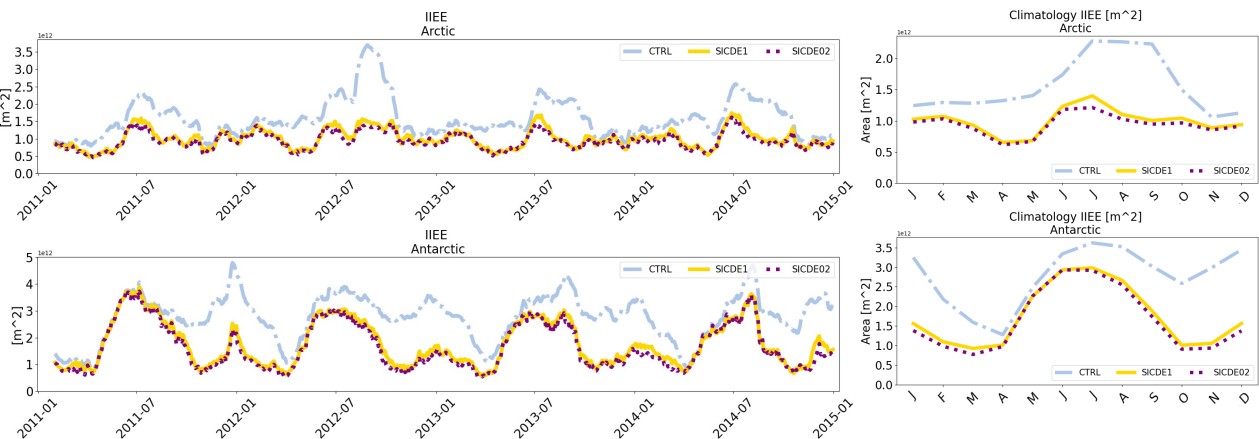

**Figure 5.** First and second row show the timeseries of IIEE for different experiments (Table 1) calculated against OSISAF data in Arctic and Antarctic regions respectively. The corresponding seasonal variability is shown in the Panels on the right.

### 4.2 Thickness observations and total sea-ice volume

The spatial SIT RMSE is calculated against the L4 CS2SMOS data, aggregating statistics from February all years (Figure 6). The BIAS maps are shown in Figure 7 with the convention: observation minus model. The CTRL experiment (Panel 6.a) shows a RMSE of 0.8 m in the Beaufort gyre and in the proximity of the Greenland coastline. Looking at the corresponding BIAS, it tends to overestimates the thickness distribution in the whole Arctic basin except for the Atlantic sector (Panel 7.a). The assimilation of SIC data (SICDE1) improves the skill-score over the Atlantic sector although no systematic impact is seen in pack-ice regions (Panel 6.b). Reducing the SIC OE (SICDE02 experiment) leads to a degradation of the thickness

RMSE and BIAS especially in the Siberian sector (Panel 6-7.c). The integration of SIT data largely improves the overall error. Cryosat-2 data (L3CR2, Panel 6.d) ameliorates the distribution in the central Arctic area (only observations higher than 0.5 m are assimilated from Cryosat-2) while no significant corrections are seen approaching the sea-ice edge that remains similar to SICDE1 experiment. The inclusion of L3 SMOS data (L3CR2&SM, Panel 6.e) shows a widespread reduction of RMSE everywhere except for the east Greenland coastline where a large positive bias of roughly 1 meter is still present. L4DE1, L4DE30, L4DESUB experiments (Panels 6-7.f,g,h), assimilate the same L4 CS2SMOS product but with different set up: 1) implementing the standard Desroziers' OE, 2) increasing the Desroziers' OE by 30 times, 3) subsampling CS2SMOS data to remove most of the off-diagonal correlations. L4DE1 shows a rather small and spatially uniform RMSE and BIAS across the basin except for the Greenland coastline where the RMSE peaks up to 0.9m at the interface between open sea and sea-ice edge. The other two experiments (L4DESUB and L4DE30) have similar skill between themselves, with larger RMSE and BIAS compared to L4DE1 close to Canadian/Greenland coastlines and in the Beaufort Gyre. A similar comparison of the November RMSE among experiments extend the validity of the present discussion to the beginning of the freezing period (not shown, see Supplement).

The timeseries of the total Arctic sea-ice volume for the different experiments are shown in Figure 8. Panel 8.a gathers mainly experiments assimilating the same dataset L4 CS2SMOS. Green crosses label values from L4 CS2SMOS weekly maps. The CTRL (blue solid line) tends to overestimate the volume of sea ice during the whole period reaching a maximum difference of $\sim 0.5 * 10^{13}$m$^3$ in March 2012 although reproducing fairly well the seasonal variability. At the onset of the freezing period, when SIT data become available in October, the sudden availability of large dense dataset generates a jump in the volume that spoils the seasonality by changing the volume minimum that usually occurs in September (L4DE1 experiment). L4DESUB produces a minor shock without changing the OE but subsampling roughly every 100km to reduce the impact of off-diagonal correlation in $\boldsymbol{R}$. The March peak is also better represented in L4DESUB rather than in L4DE30, where instead a multiplicative factor of 30 is applied to OE. Such multiplicative factor could be reduced to have a worse but acceptable jump in October and a better peak in March, in order to match the skill-score of L4DESUB. Experiment L4DE provides the best initial conditions for forecasting purpose however at the cost of losing the consistency with past timeseries. The use of subsampling scheme is able to preserve the seasonality and can be instead considered for Reanalysis purpose.

Panel 8.b highlights the effect of assimilating different datasets. The SICDE1 experiment positioned the minimum of volume in September but has little impact on the rest of the timeseries, correcting the thickness field only close to sea-ice edge. The assimilation of Cryosat-2 data (L3CR2 experiment) reduces the volume overestimation that is however still present especially in March. Adding the assimilation of thin ice (L3CR2&SM experiment), the total volume is much better represented together with its seasonality.

### 4.3 Observation influence

A measure of the relative influence of different observation types into the model dynamic and thermodynamics, follows the evaluation of the Degrees of Freedom for Signal (DFS) established in Cardinali et al. (2004). DFS is defined as the trace of the derivative of the analysis with respect to the observations in the observation space. DFS measures the sensitivity of the model

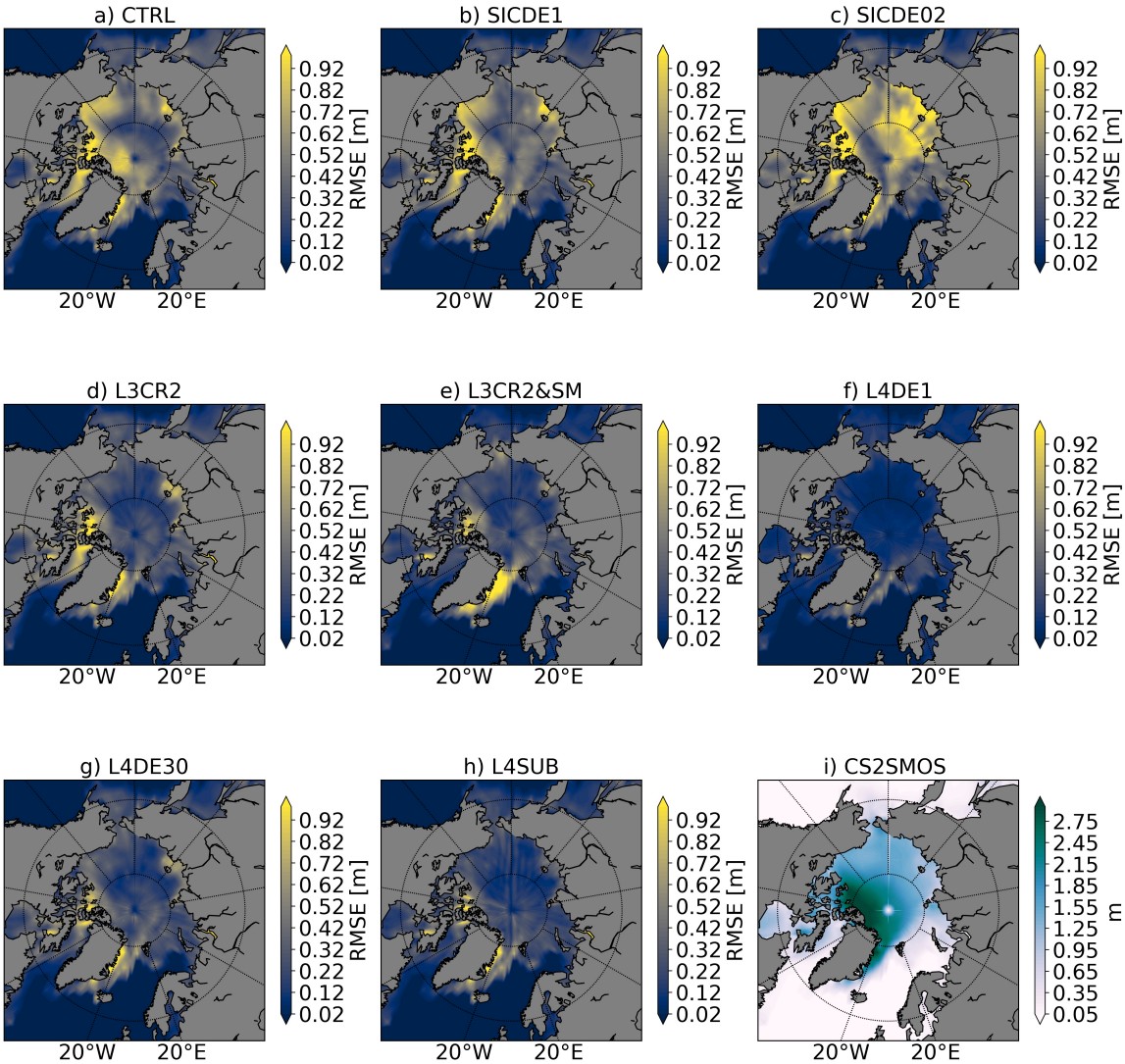

**Figure 6.** Spatial thickness RMSE for different experiments (Table 1) calculated aggregating the February statistics for all the years

to the observation variation and is able to leverage different types of observations, quantifying the relative impact of each single dataset:

$$\text{DFS} = \text{Tr}\left\{\frac{\delta(\boldsymbol{H}\boldsymbol{x}_a)}{\delta\boldsymbol{y}}\right\} = \text{Tr}\{\boldsymbol{H}\boldsymbol{K}\} = (\tilde{\boldsymbol{y}} - \boldsymbol{y})\boldsymbol{R}^{-1}\boldsymbol{H}(\tilde{\boldsymbol{x}}_a - \boldsymbol{x}_a) \tag{5}$$

where $\boldsymbol{K}$ is the Kalman gain, $\boldsymbol{H}$ is the observation operator that projects the analysis in the observation space while $\tilde{\boldsymbol{y}}, \tilde{\boldsymbol{x}}_a$ denotes the perturbed observations and the corresponding analysis. In practice, DFS can be computed with a randomisation technique (Chapnik et al., 2006) and it is commonly applied to a 3dvar framework by averaging over the number of observations

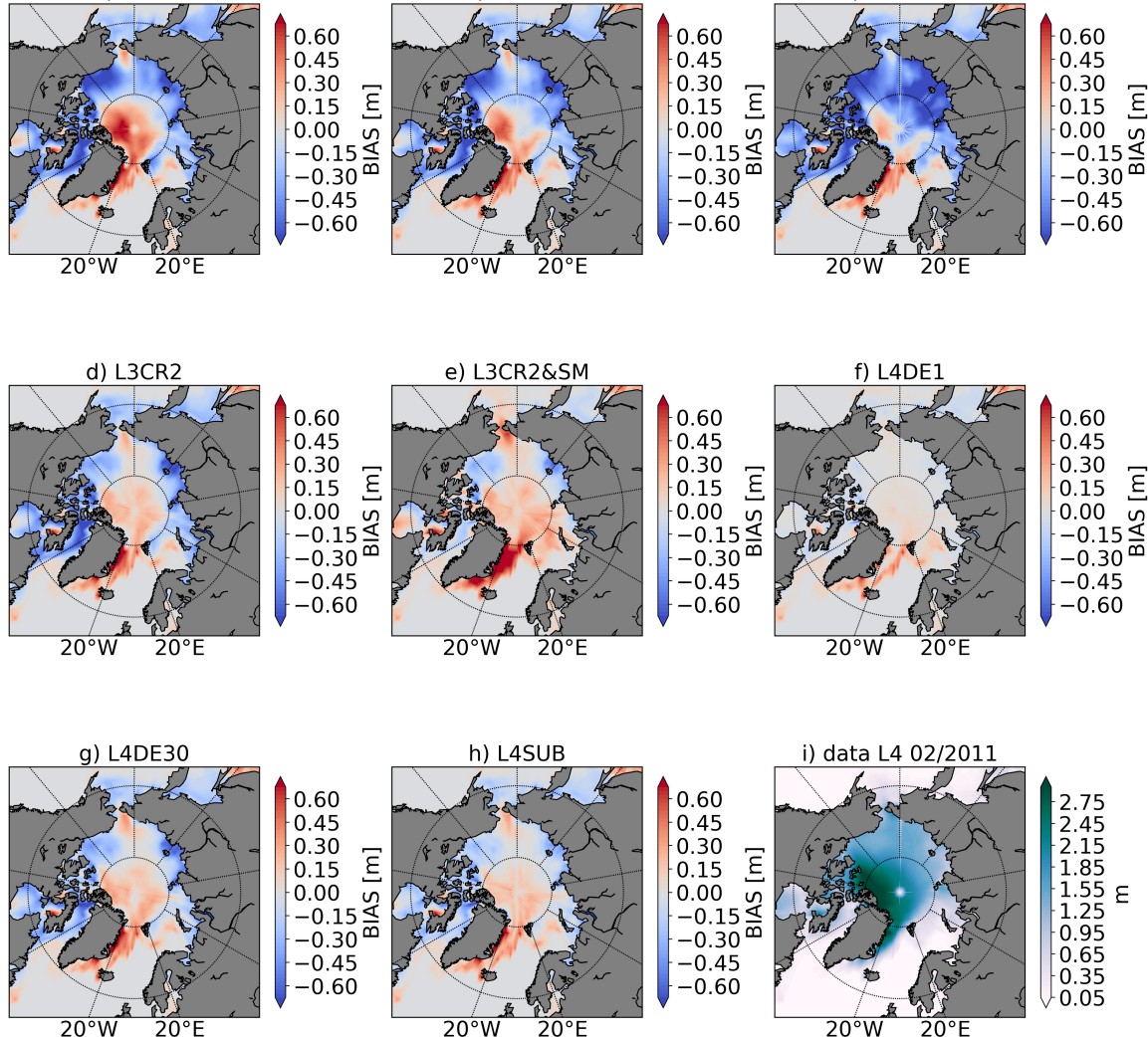

**Figure 7.** Spatial thickness BIAS (observation minus model) for different experiments (Table 1) calculated aggregating the February statistics for all the years

$\overline{\text{DFS}}$ (Montmerle et al., 2007; Storto and Thomas Tveter, 2009; Storto et al., 2010). In Xie et al. (2016, 2018), $\overline{\text{DFS}}$ is used to
compare the influence of different observation datasets by defining a relative DFS (RDFS) or Impact Factor (IF):

$$\text{IF}_j = \frac{\overline{\text{DFS}_j}}{\sum_o \overline{\text{DFS}_o}} \qquad (6)$$

with $o$ running over different instruments or datasets. In practice, $\text{IF}_j$ quantifies the importance of $j$-th dataset compared to the others. Perturbations were generated from a Gaussian distribution with zero mean and imposing the observation error as standard deviation. Figure 9 shows the spatial IF in L4DE1 and L3CR2&SM experiments, calculated over the period November

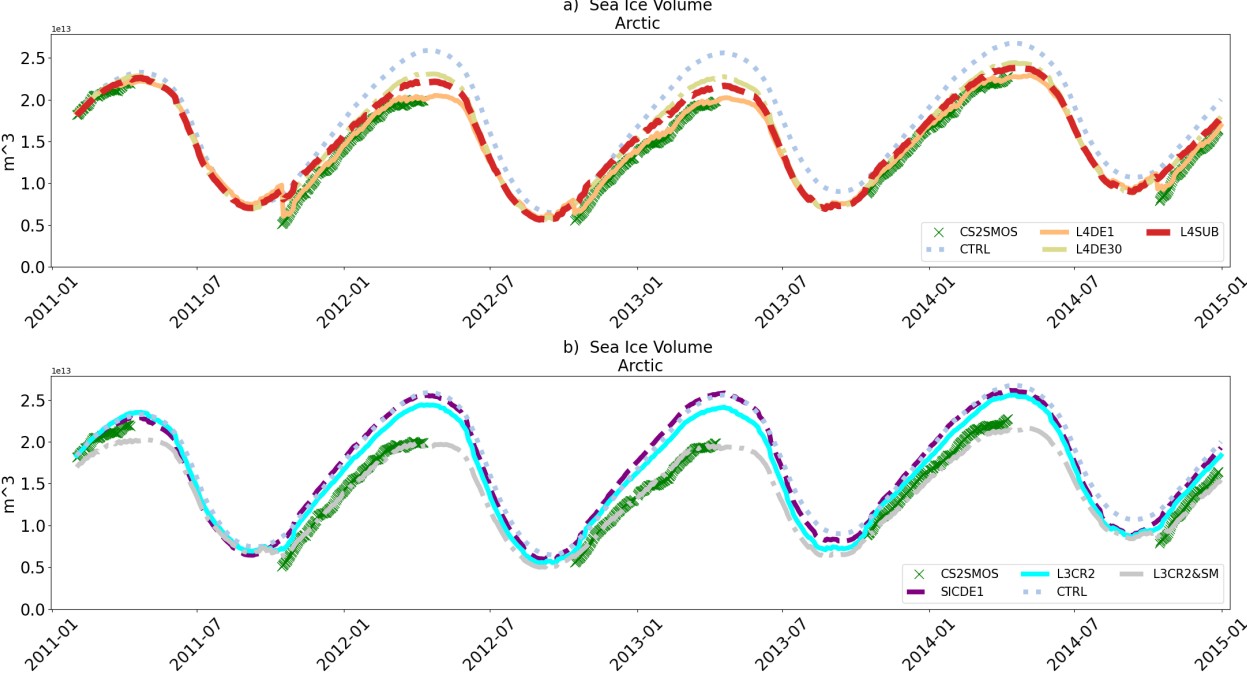

**Figure 8.** Panel a shows the timeseries of the total sea-ice volume in the Arctic for several experiments with different DA set up against observation estimates from L4 CS2SMOS data. Same thing for Panel b where the impact of the assimilation of different SIT datasets is highlighted.

2012-February 2013. SIC data generally show little influence on the central Arctic area where sea-ice is fully packed with concentration close to 1. As we move towards the sea ice edge, the impact reverses and SIC influence rapidly saturates at 1 (L4DE1 eperiment). This sharp discontinuity is likely to come from an overestimation of the SIT error compared to the SIC one. In the L3CR2&SM experiment (Figure 9.b), we can discriminate the influence of the two independent SIT datasets. Cryosat-2 data largely impact the Eurasian basin where the thickness is usually higher than 0.5m. Most the Siberian coast is instead driven by the SMOS data as well as west Greenland rift basin. Moving toward the sea-ice edge a competitive behaviour is shown between SMOS and SIC data: the two datasets almost equally contribute to modify the model thermodynamics.

## 4.4 Validation against BGEP mooring data

An independent validation can be carried out thanks to the Beaufort Gyre Exploration Project (BGEP, www.whoi.edu/beaufortgyre) from the Woods Hole Oceanographic Institution. BGEP provides high-frequency data of sea-ice drafts from moored sonars (Krishfield et al., 2014) in three different positions of the Beaufort Gyre that slightly change year by year: mooring A located approximately at $\sim [75°N;154°E]$, mooring B at $\sim[78°N;150°E]$ and mooring D at $\sim [74°N;139°E]$. Sea-ice draft measurements are transformed into thickness estimates using a simple multiplicative factor of 1.1 as in Mu et al. (2018a) being representative

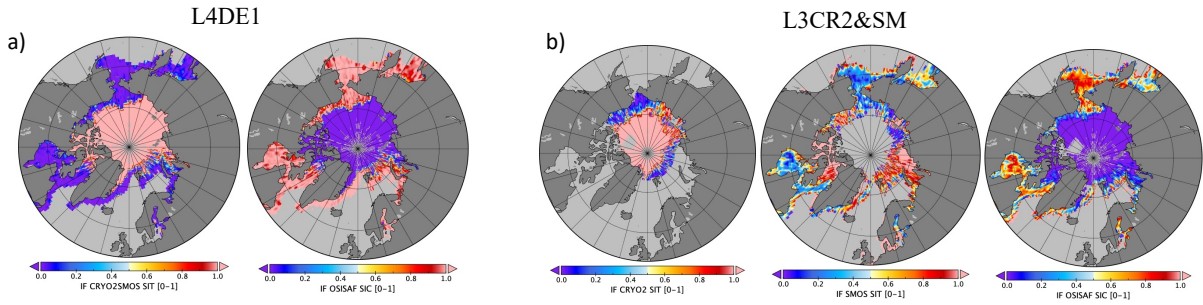

**Figure 9.** Spatial IF for L4DE1 (Panel a) and L3CR2&SM (Panel b) experiments. Panel a shows the relative influence/strength of CS2SMOS data and OSISAF one. Panel b considers the same for L3 Cryosat-2, L3 SMOS and OSISAF data

of the ratio between the mean seawater and sea ice density (Nguyen et al., 2011). A more sophisticated approach considers a balanced equation that implies the knowledge of the snow depth that is usually extracted from an ensemble of simulations
(Xie et al., 2018; Alexandrov et al., 2010), thus being influenced by the specific set-up of the ice model itself. In the following, we prefer to use the first approach being totally model independent. Figures 10-11 show the timeseries of sea-ice thickness for different experiments compared to the mooring measurements and estimates from L4 CS2SMOS. CS2SMOS maps represent generally well the trends during the freezing season, although some discrepancies can be found at the end of 2012 where it overestimates the thickness at position A and B by 0.7m.

During the melting season, the CTRL experiment predicts on average 1.5 m of ice at the three positions, always overestimating the observations. The assimilation of SIC (experiment SICDE1) is able to reduce such overestimation at the position A during the summer months, while less impact is seen at mooring B and D. The assimilation of CS2SMOS maps (L4DE1,L4DESUB) yields the model thickness to be much closer to mooring measurements: in winter the BIAS almost disappears, while during summer the RMSE is reduced below 0.5 m in all the three positions. L4DE1 experiment closely follows the
evolution of CS2SMOS data thus generating a strong discontinuity at the beginning of the fall season of 2011 that is instead not present in the experiment L4DESUB. The assimilation of SIT data in winter (experiments L4DE1 and L4DESUB) provides much better initial conditions for SIT prediction in spring compared to experiments without SIT assimilation. SIT estimates at the onset of the next fall season is also better predicted in the Beaufort Gyre. Figure 11 groups experiments that uses different thickness datasets. Cryosat-2 data (L3CR2 experiment) reduce the BIAS over the whole time-series compared to CTRL run.
The addition of SMOS data (L3CR2&SM experiment) bring SIT values closer to the observations and similar to L4DE1 skill score (assimilation of CS2SMOS maps). L3CR2&SM seems to correct the overestimation of 0.7 m present in L4DE1 during winter 2012-2013 at position D, although discontinuties can be spotted in some cases when thin SIT data (SMOS) become available.

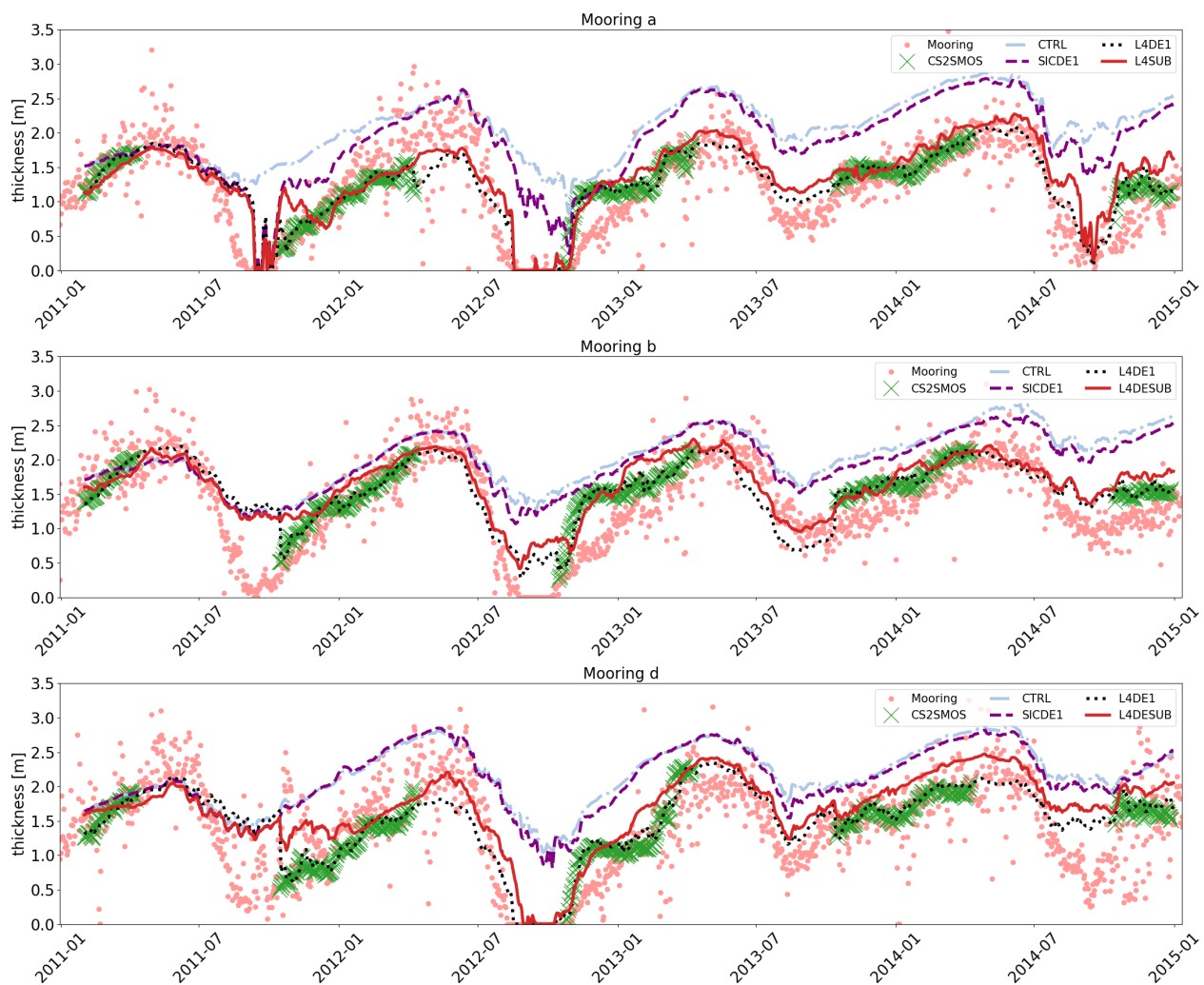

**Figure 10.** Timeseries of thickness values from several experiments (Table 1) with different DA set up at three positions in the Beaufort Gyre (mooring A, B and D, see text). Solid lines label different experiments, pink dots refer to daily data measured by the Beaufort Gyre Exploration Project while green crosses are estimates from L4 CS2SMOS maps.

## 4.5 Validation against Operation IceBridge data

A second independent dataset is available for the same period, gathering data from several campaigns of airborne surveys on polar ice thanks to the NASA Operation IceBridge project (https://www.nasa.gov/mission_pages/icebridge/). Different instruments were installed on board of the aircrafts from Snow Radars to Airborne Topographic Mappers, providing sea-ice freeboard, snow depth, and sea-ice thickness measurements (Kurtz et al., 2013). Specifically, in the present exercise, we used the IceBridge L4 Sea Ice Freeboard, Snow Depth, and Thickness (IDCSI4), Version 1, http://nsidc.org/data/idcsi4 (Kurtz et al.,

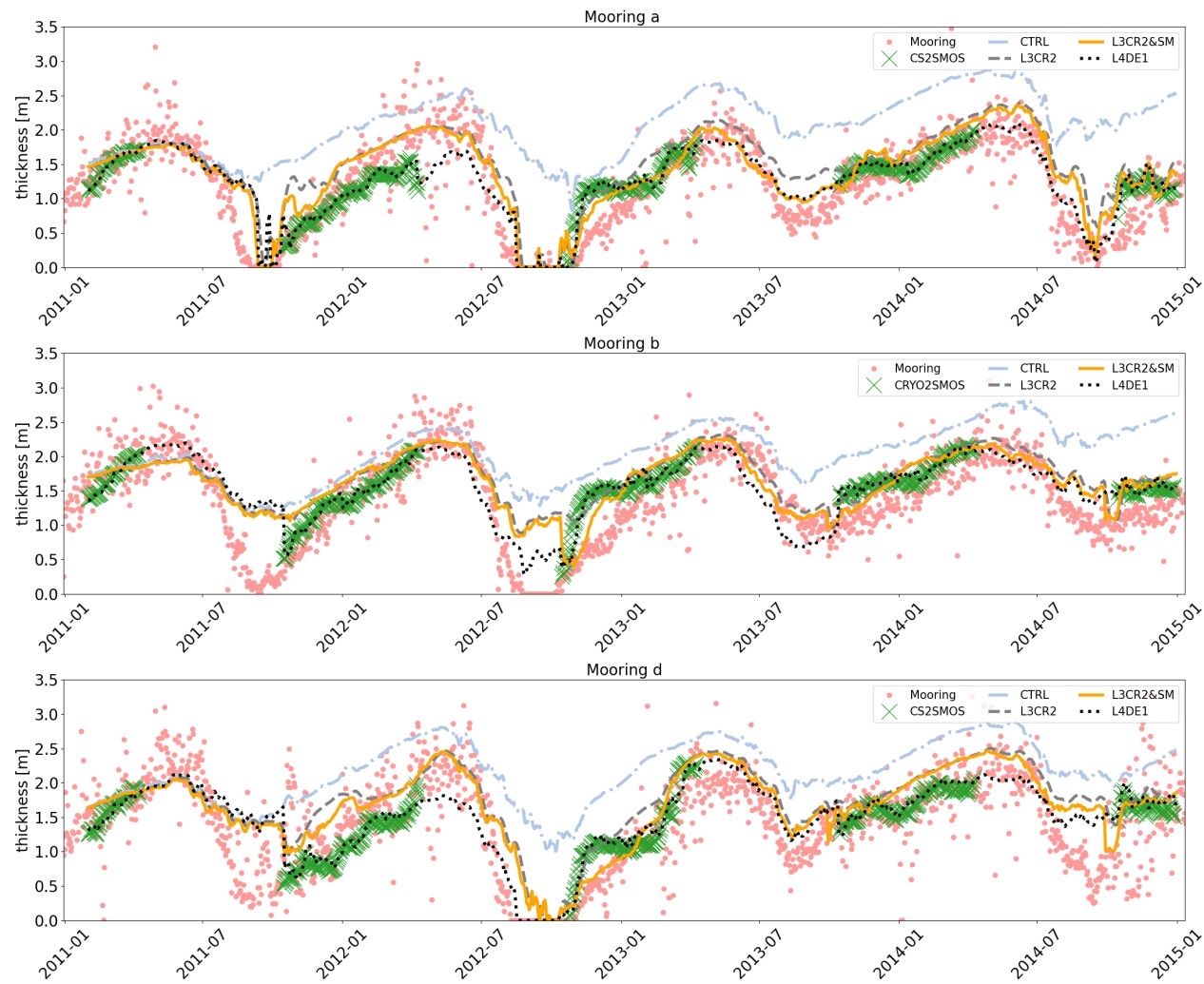

**Figure 11.** Timeseries of thickness values from several experiments (Table 1) with different set of observations assimilated at three positions in the Beaufort Gyre (mooring A, B and D, see text). Solid lines label different experiments, pink dots refer to daily data measured by the Beaufort Gyre Exploration Project while green crosses are estimates from L4 CS2SMOS maps.

2015) between 2011-2013 since no data are available for 2014. Such dataset covers several days in March and April when satellite SIT retrievals are no longer available and no SIT assimilation is performed. Results are summarised in Figures 12-13 , containing the SIT spatial RMSE and BIAS for different experiments and binned in $2°x2°$ boxes. The conclusions discussed in the BGEP section are confirmed and extended to a broader region. The differences in the skill scores among the experiments largely depend on the diverse initial conditions in mid-March. Winter assimilation of SIT data (panels d-h) produces a smaller RMSE in March-April statistics compared to SIC-only (panel b-c) and CTRL experiment (panel a). A spatial dipole structure for BIAS (observation minus model) is generally seen in all the experiments with an overestimation of thickness in the Beau-

fort Gyre and an underestimation in the Atlantic sector. The L4DE1 experiment (assimilation of CS2SMOS with Desroziers' error) shows the lowest RMSE and reduces the regional BIAS almost everywhere. SICDE02 (assimilation of SIC with reduced observation error) shows the worst skill in term of regional RMSE and BIAS. However negative/positive BIASES seem to compensate each other producing the lowest global BIAS (spatially averaged). This demonstrates that such indicator is not always representative of the actual skill of the model. Subsampling the data (L4SUB, panel h) or increasing the observation errors (L4DE30, panel g) still provide positive feedback in April.

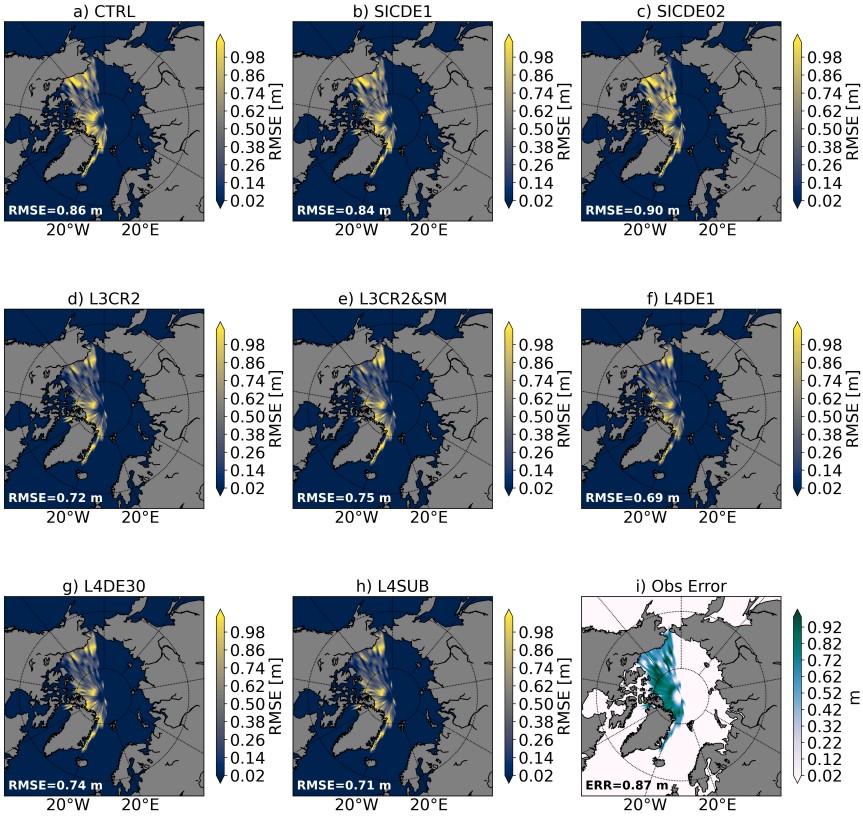

**Figure 12.** Spatial thickness RMSE for different experiments (Table 1) calculated against Operation IceBridge SIT measurements, aggregating the March-April statistics for all the years and binned with $2°x2°$ boxes

## 5  Conclusions

Despite the availability of different types of sea-ice observations in the last decade, their joint assimilation in a multivariate framework is still an active research field. Sea-ice variables generally follow a bounded distribution that can peak over one of the two boundary values. Thickness measurements show limited accuracy (Zygmuntowska et al., 2014) with CryoSat-2 data providing high signal-to-noise ratio only for thick sea-ice, while SMOS data for thin one. Recently, Ricker et al. (2017)

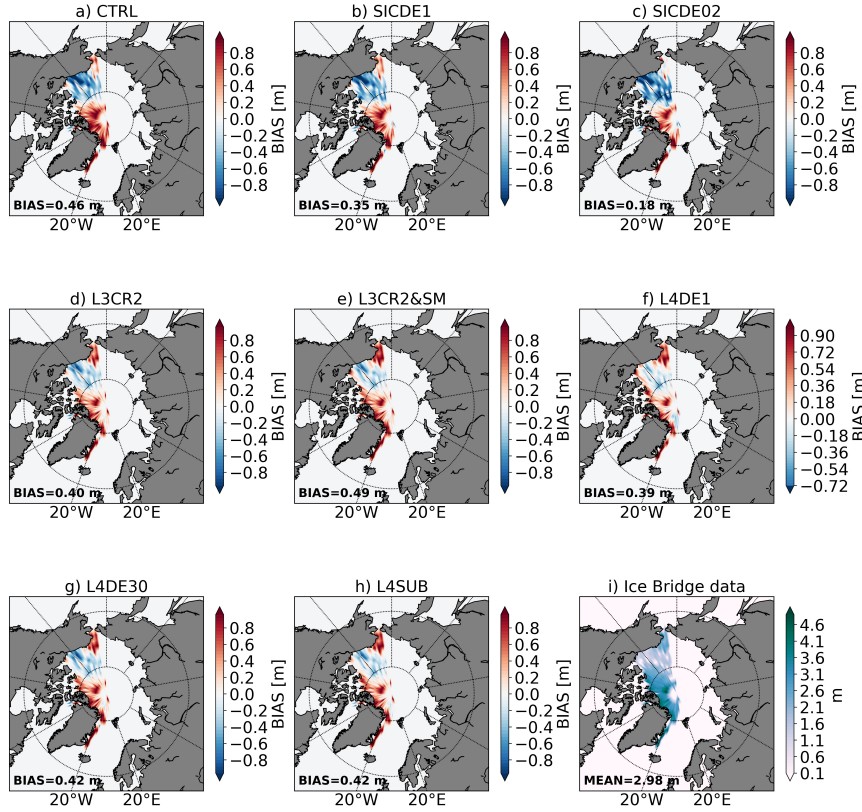

**Figure 13.** Spatial thickness BIAS (observation minus model) for different experiments (Table 1) calculated against Operation IceBridge SIT measurements, aggregating the March-April statistics for all the years and binned with 2°x2° boxes

showed that such datasets are complementary and can be merged yielding to an optimally-interpolated spatially-reconstructed thickness distribution CS2SMOS. However the straightforward assimiltion of such maps can produce discontinuities in the sea-ice volume at the onset of the accretion period whether the observation errors are not properly tuned, thus spoiling the seasonal variability.

In this study we extend a 3DVar scheme, called OceanVar, employed in the routine production of CMCC global/regional analysis/reanalysis, to benefit from sea-ice concentration and thickness. Those variables are treated through an anamorphosis operator that is included in the control vector transformation composing the $B$ matrix. Such operator transforms the probability density functions of sea-ice anomalies into Gaussian ones theoretically without loss of information (Bertino et al., 2003; Brankart et al., 2012), being more adeguate to treat non linear dependencies (Corder and Foreman, 2009). We showed that such transformation is also able to preserve the strong anisotropy of sea-ice fields close to sea-ice edge, thus helping future coupling with ocean variables.

A set of global ocean/sea-ice experiments are performed for a period of four years with different DA setup and assimilating different observation datasets. The sole assimilation of SIC data provides a positive but small improvement in the representation of thickness field that can be potentially degraded in the case that the error assigned to SIC data is too small. The model thickness field starts matching the observed one only when Cryosat-2 data are employed while the addition of SMOS data further reduces the volume overestimation by constraining the thin sea-ice especially close to the edge. The intermittent availability of SIT data along the year, together with the lack of off-diagonal elements in the $R$ matrix, can generate jumps in the total volume that can spoil the seasonal variability and requires extra tuning. Through the analysis of Degrees of Freedom for Signal (Cardinali et al., 2004), the relative influence of different types of observations is also studied showing the competitive behaviour of SMOS and OSISAF data for thin ice.

Two independent validations are carried out against mooring data in the Beaufort gyre (Beaufort Gyre Exploration Project) and sea-ice thickness measurements from NASA Operation IceBridge project. The assimilation of merged product CS2SMOS and the joint assimilation of L3 Cryosat-2 and SMOS data provide similar skill scores. These two configurations outperform the other set up during the melting period, where no satellite thickness data are available, demonstrating that the benefits of realistic initial conditions in the Beaufort gyre can last up to 6-month at least.

*Data availability.* All the sea ice reanalysis experiments are available on request. Sea ice concentration data were downloaded from EUMETSAT Ocean and Sea Ice Satellite Application Facility, Global sea ice concentration climate data record 1979-2015 (v2.0, 2017), OSI-450. Data extracted from OSI SAF FTP server/EUMETSAT Data Center: (2011-2015) (global) accessed June 2019 (OSISAF, 2021). SMOS and Cryosat-2 products were downloaded from www.meereisportal.de/ portal (Grosfeld et al., 2016). The production of the merged CryoSat-SMOS sea ice thickness data was funded by the ESA project SMOS & CryoSat-2 Sea Ice Data Product Processing and Dissemination Service, and data from 2011 to 2015 were obtained from Alfred Wegener Institute (AWI). Independent validation is performed against: i) data collected and made available by the Beaufort Gyre Exploration Program based at the Woods Hole Oceanographic Institution (https://www2.whoi.edu/site/beaufortgyre/) in collaboration with researchers from Fisheries and Oceans Canada at the Institute of Ocean Sciences; ii) Data from NASA Operation IceBridge, specifically the IceBridge L4 Sea Ice Freeboard, Snow Depth, and Thickness (IDCSI4) data set, Version 1, http://nsidc.org/data/idcsi4 (Kurtz et al., 2015)

*Author contributions.* A.C. designed and conducted the experiments. D.S.B. contributed to the experiment design. D.S.B. and A.Y. provided expertise on the data assimilation while D.I. on sea-ice modelling. D.I. and S.M. contributed to the result interpretation. A.C. lead the writing of the first draft that was modified and corrected by all the Authors

*Competing interests.* The authors declare that they have no conflict of interest

*Acknowledgements.* The activities leading to these results have been contracted by Mercator Ocean International under GLORAN project, that implements Copernicus Marine Environment Monitoring Service (CMEMS) as part of the Copernicus Programme. Dr. Andrea Storto (CNR) is thanked for enlightenment discussion on the topic.

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
