# Peer review of "Bivariate sea-ice assimilation for global ocean Analysis/Reanalysis"

_EGUsphere, 2023_

## Author Comment (AC1)

**Reviewer 1 : Comments and Answers**

In this study, a coupled ocean-sea ice 3Dvar system is extended to include the assimilation of sea ice concentration and sea ice thickness. OSISAF sea ice concentrations and sources of sea ice thickness from CryoSat-2 and SMOS are assimilated in various configurations. The assimilation of OSISAF sic data alongside L4 C2SMOS SIT data with a Desroziers' OE factor of 1 performs best in comparison to both assimilated and independent moorings.

Overall I believe this to be a good paper with strong scientific basis and quality, particularly with strong implementation of a robust data assimilation scheme and good statistical assessment of the results. The plots show the results well. My main concern is that a bit more validation would improve the impact of the paper strongly alongside some discussion of the results.

We thank the Reviewer and we do agree that a second independent validation can improve the quality of the manuscript. Following the recommendation of both Reviewers we enriched the text adding a new validation dataset and discussed the corresponding metrics. The abstract is also reworded and corrected as suggested.

**General comments:**

In your analysis of the validation against BGEP ULS moorings you also mention the RMSE and BIAS but do not show these results which would be useful to see. I think it would be useful to extend the validation to also look at Operation IceBridge data, which also covers the time period of your experiments, and has a higher spatial coverage than the BGEP ULS moorings do. Although the data only cover March and April it would be useful to show a comparison to them with, for example, BIAS, RMSE or scatter plots of the different runs against the OIB data available during your experimental time period.

Operation IceBridge measurements are now used as a second independent validation dataset and a new section is added. Specifically we use the IDCSI4 dataset version 1 (http://nsidc.org/data/idcsi4; Kurtz et al., 2015) extracting data from 2011 to 2013 (no data available in 2014). Results are summarized in Figures S1 and S2 below, containing the SIT spatial RMSE and BIAS respectively for different experiments and binned in 2°x2° boxes. Metrics are calculated aggregating statistics from late-March and April (the only months available in the datasets for those years). Such dataset covers several days after the end of the dissemination of satellite data therefore the differences in the SIT distributions among experiments largely depends on the diverse initial conditions from mid-March. Figures S1 and S2 confirm the conclusions discussed with the analysis of BGEP ULS data and extend them to a broader region. Winter assimilation of SIT data (panels d-h) produces a smaller RMSE in March-April statistics compared to SIC-only (panel b-c) or CTRL experiment (panel a). A spatial dipole structure for BIAS (observation minus model) is generally seen in all the experiments with an overestimation of thickness in the Beaufort Gyre and an underestimation in the Atlantic sector. L4DE1 experiment (assimilation of CS2SMOS with Desroziers' error) shows the lowest RMSE and reduces the regional BIAS almost everywhere. SICDE02 (assimilation of SIC with reduced observation error) shows the worst skill in term of regional RMSE and BIAS. However negative/positive BIASES seem to compensate each other leading to a low global BIAS (spatially averaged). This demonstrates that such indicator is not always representative of the actual skill of the model. Subsampling the data (L4SUB, panel h) or increasing the observation errors (L4DE30, panel g) still provide positive feedback in April.

The distribution of observation errors as provided by Operation IceBridge is also shown in panel i) for comparison.

---

## Author Comment (AC2)

**Reviewer 2 : Comments and Answers**

The paper presents experiments where sea ice concentration and thickness observations are assimilated in a coupled ice-ocean model. The correlation between sea ice concentration and thickness observations is taken into account. The method of Desroziers (2005) is used to better characterize the observation-error statistics. Also the Degrees of Freedom for Signal (DFS) developed by Cardinali et al. (2004) is used to measure the influence of the observations on the analysis.

The results of the experiments are first compared to one of the same sea ice observations dataset that were assimilated. Since only the analyses and not the forecasts are verified, the exercise is more to verify the assimilation methodology than the accuracy of the analyses. Fortunately, the experiments are also verified against independent observations of ice thickness in the Beaufort Sea.

We thank the Reviewer, and we agree that the quality of the Manuscript can increase expanding the validation section. Following a similar comment from the Reviewer 1, we add a second independent comparison against measurements from NASA Operation IceBridge project (http://nsidc.org/data/idcsi4 ; Kurtz et al., 2015) that cover several regions of the Western Arctic for several days in March and April between 2011 and 2013. The results are summarized in the initial answer to the first Reviewer (please see figure S1 and S2) and confirm the conclusions coming from the comparison against BGEP ULS moorings. Winter assimilation of SIT data produces a smaller RMSE in March-April statistics compared to SIC-only or CTRL experiment (no assimilation). Among the different experiments with SIT assimilation, the L4DE1 run (assimilation of CS2SMOS with Desroziers' error) shows the lowest RMSE and reduces the regional BIAS almost everywhere.

Kurtz, N., Studinger, M., Harbeck, J., DePaul Onana, V., and Yi, D.: IceBridge L4 Sea Ice Freeboard, Snow Depth, and Thickness, Version 1, https://doi.org/10.5067/G519SHCKWQV6, 2015.

In the section describing the assimilation method, we see gICE and gSIC that are used interchangeably. I think it would be better to have just one expression throughout, if these point to the same thing.

The difference between gICE and gSIC was not explained, we thank the Reviewer and clarify in the text. While gSIC and gSIT refer specifically to the corresponding gaussian counterpart of SIC and SIT, gICE is used in a more general way, $V_{gICE->ICE}$ points to a generic mapping of the sea-ice variable from Gaussian space to the physical one and can be associated separately to both SIC and SIT.

Figures 10 and 11 seem to be identical. Please remove figure 11 and only refer to figure 10 in the text. Also please revise the caption for this figure. Currently the caption for figure 10 and 11 are slightly different but I am not sure which one is more correct.

Figures 10 and 11 gather results from two different subsets of experiments from Table 1 to highlight different aspects. Fig 10 compares the sole assimilation of SIC with the experiments where both SIC and SIT are used, in particular, L4DE1 experiment (assimilation of the merged L4 product) and its subsampled version (L4DESUB). Fig 11 groups experiments that uses

different thickness datasets where the impact of the sole assimilation of the CryoSat-2 and the impact of the independent assimilation of CryoSat-2 and SMOS data are compared to the L4DE1 experiments. Anyway, we agree that the color scheme can confuse the Reader, therefore, we changed the figures to avoid the use of the same color for different experiments

**Specific comments:**

Line 20: What about the first CryoSat mission ?

The very first satellite associated with the CryoSat mission was lost due to a failure during its launch in 2008. The second launch was successful, and it is common to refer to the second mission of CryoSat as CryoSat-2.

Line 65: "contest" or "context" ?

Context is the correct word, thank you.

Line 88: 'x' should be called the state vector, and '$x_a$' is the final analysis state (the value of 'x' that minimize the cost function).

We rephrase the line, "$\delta x$ label the increments that correspond to the difference between the final analysis state $x_a$ and the initial ocean state $x_b$, in the minimum of the cost function." The definition of x as the state vector comes a few lines below.

Line 113: Please define "CVT".

The acronym was indeed not defined, now we added "Control Vector Transformation or CVT", thanks

Line 146: Should "$V_{ICE \rightarrow gICE}$" be "$V^T_{ICE \rightarrow gICE}$" ?

Thank you, gICE was misplaced, we correct the initial "$V_{ICE\text{->}gICE}$" to "$V_{gICE \rightarrow ICE}$"

Line 151: I think that "that it has been shown" should be "that has been shown".

Corrected, Thanks

**Technical corrections**

Line 122: "31-year-long" should be "31-year long" for consistency with the caption of figure 1.

We uniform text and caption to "31 year-long" that means a single simulation lasting 31 years.

Figure 1: In the caption, "Panel g and f show the same as e,f for September" should be "Panels g and h show the same as e,f for September".

Figure 4: In the caption, "Panels" should be "panels".

Line 203: "Beaufourt" should be "Beaufort".

Line 219: "0.5 * $10^{13}$km" should be "0.5 * $10^{13}$ $m^3$"

Line 253: "is usual higher" should be "is usually higher".

Line 255: "…contribute to the modify the model…" should be "…contribute to modify the model…".

Line 275: "indestion" should be "ingestion".

We thank you the Reviewer for the corrections. We applied all in the revised version of the manuscript.

**Citation**: https://doi.org/10.5194/egusphere-2023-254-RC

---

## Author Comment (AC3)

**Reviewer 3 : Comments and Answers**

I have asked these questions orally on the IICWG workshop, but here is a written version. The paper by Cipollone and colleagues is to my knowledge the first application of the anamorphosis to the assimilation of sea ice variables going through several assimilation cycles. The work carried out is of very high quality and the results are quite encouraging but the paper would deserve a few clarifications before publication.

A Gaussian anamorphosis is a continuous function and is therefore not designed to turn discontinuous probability densities (like the zeroes and 100% of ice concentrations in open water and fully packed ice). In case of accumulation of probability density at a given value ("atoms of probability density"), the piecewise linear mapping to 21 quantiles will diffuse the atoms to neighbouyring values. The authors should explain how both extremities of the distribution are handled.

We thank Dr. Bertino for reading and commenting the Manuscript pointing out the aspects to be explained in more detail. We extended the description the anamorphosis operator (called Mapping below) to clarify how it is employed in the DA system, by including the optimal range of application and possible limitations. The present DA system uses the tangent/adjoint version of the Mapping linearized around the background value $V_{gSIC\text{->}SIC} = [\frac{\partial MAP}{\partial SIC}]_{SIC=SIC_B}$. The derivative is a simple numerical centered first-ordered difference around the background, except for the extreme where it is a backward or forward formulation. The existence of $V_{gSIC\text{->}SIC}$ requires the mapping to be locally continuous and the diffusion towards the neighboring values help in this sense; in the case the derivative does not exist the corresponding increment is zero. We tend to avoid the presence of discontinuous probability densities that reflect an underestimation of model error (under the assumption that the variability of the model reflects its error, i.e., zero standard deviation). To avoid such underestimation, we augment the number of model samples in each point by adding values from neighboring points to construct a transformation that could span all possible physical values. A second possible approach can be the use of values from previous or next month.

Further, the paper does not explain how values out of bounds are treated. If the model forecast produces a sea ice thikness value larger than the largest of the samples, what will be its Gaussian counterpart? In Simon and Bertino 2011, we extrapolated the last piece of the linear mapping of quantile according to an exponential tail (Eq 15). This could be included to avoid trouble.

We added a description concerning the treatment of extreme values. There are two different types of possible extreme values: either in the observations or in the background. Extreme values in the observations, i.e. observations far from the background, can be assimilated with current system. However, the tangent linear approximation is suboptimal because implicitly supposes that final increments do not diverge much from the background, the coefficients in the $V_{gICE\text{->}ICE}$ are valid around the background. This is a common problem of the tangent linear approximation and of variational DA that are not designed for the assimilation of extreme values. The use of background quality checks in the preprocessing serves to remove values that are far from the background and for which the linear approximation does not hold anymore.

In the case that the values out of bounds are in the forecasts, i.e., out of the range of values extracted from the historical simulation, then it is not possible to calculate the derivative of the

Mapping and the operator reduces the increments to zero. We preferred to stay conservative and not correct such extreme events that will be driven only by the model. The idea of extrapolating the distribution can be a solution in case a correction is needed. Probably the best approach would be the use of a hybrid scheme, with a part of the B coming from an ensemble that goes to: i) add the model "error-of-the-day"; ii) update the Mapping with the inclusion of ensemble forecasted values.

We added the following paragraph:

"It is worth to note that the use of tangent/adjoint approximations of the anamorphosis leads the assimilation of extreme values, to be suboptimal (i.e. observations that are far from the background value). Tangent/adjoint approximations of any operator are valid in the proximity of the background value and become less and less accurate in the case of large corrections and highly non-linear operator. This is anyway a limitation that is implicit in any three-dimensional variational scheme. Moreover, the anamorphosis should span all the possible physical values in each grid point. In the case the background is out of the range of values used for the anamorphosis, then it is not possible to calculate the derivative and the corresponding increments are zero. This means that extreme events in the background (not present in the 31 years of simulation) do not receive correction. In Simon and Bertino (2012) they include an exponential tail to the anamorphosis, in order to treat values out of bounds. A further approach could be the use of a hybrid **B** where the ensemble part goes to update the anamorphosis with the inclusion of new model values as well as adding the "error-of-the-day".

Besides these two remarks, some clarifications could be made regarding the algorithm:

- Abstract: "transform sea ice anomalies into Gaussian control variables". Why anomalies and not the full field values?

Following also the comment of the first Reviewer we rephrase the abstract. In the specific case, the sea-ice anomalies that were transformed in the control variables (using the tangent/adjoint operators) refer to $\delta x$ that can be considered anomalies with respect to the background in a statistical sense. The operators used are however the tangent and adjoint version of the Mapping not the full operator. The phrase is changed accordingly:
"The tangent/adjoint versions of an anamorphosis operator are used to transform locally the sea-ice anomalies into Gaussian control variables and back, minimising in the Gaussian one."

- l40: the result of the anamoprhosis is not strictly Gaussian. It would be fair to write "more Gaussian" or "closer to a Gaussian".

We change the sentence in "The operator transforms the probability density functions of SIC/SIT anomalies towards Gaussian-like ones performing the minimization in this space".

- Section 3: linearizing the anamorphosis operator seems to defeat the purpose of the anamorphosis. Can you clarify why and what is done there in practise with a piecewise linear quantile mapping?

The linearization consists of a numerical derivative of the quantile mapping in each grid point around the background value. This is the classical approach where the full Mapping is replaced locally with the first two terms of the Taylor expansion: the approximation holds if the increments are not far from the background value. Using a local Gaussian space in each point

of the grid is optimal for a correct application of the rest of the Control Variable Transformation, i.e., the cross-correlation and horizontal diffusion. The latter mimics the Gaussian spread of information in the surrounding points based on a reference-length using three iterations of a first-order recursive filter. The benefit turns to be significative for example close to sea-ice edge. In the physical space, correlating two points that have opposite distributions (say being close to 100% of SIC in one point and close to 0% in the other) can populate the surrounding points with values that do not fall in the range of their distributions. The use of a gaussian space re-center the increments in the range of physical values. In this sense this local mapping easily allows the use of diverse correlation length for each grid point, as the maps provided for example by CS2SMOS. Moreover, such operator helps the future coupling with Gaussian-like ocean variables such as temperature and salinity. Without such Mapping, the isotropic spread of temperature or salinity increments on the edge of sea-ice, would lead to a corresponding spread in SIC and SIT, potentially destroy, or smooth the edge of sea ice. We rephrase the corresponding paragraph to be clearer:

"The use of local Gaussian space in each point of the grid turns out to be crucial for a correct application of the horizontal correlation operator especially close to sea-ice edge. Figure 2 shows the sea-ice increments in a test case, says the third week of February 2015, generated with and without the application of $V_{gICE \rightarrow ICE}$ with a large fixed correlation length of 150km and three iterations of a first order recursive filter. Green solid line corresponds to the mean sea-ice edge in that week, SIC and SIT are jointly assimilated close to the sea-ice edge. In the physical space an isotropic spread of information towards the ice-free areas is seen (Figures 2.c,d). The use of $V_{gICE \rightarrow ICE}$ "re-center" the increments (in the Gaussian space) within the range of physical values, reducing the wrong isotropic diffusion (Figures 2.a,b) and following the variability of the specific region"

- If V gICE->ICE and V^T ICE->gICE are linearized anamorphosis functions, where are the nonlinear anamorphosis functions used? If they are both nonlinear they must be the forward and backward anamorphosis functions, please clarify.

Being the mapping empirical, the linearization is numerical around the background value. The code reads the full mapping, reads the background state value, and compute the derivative with a simple first-order central difference around the background value.

- l.87: isn't it conditioned to the model background rather than analysis?

Thank you for the correction, the text is changed accordingly.

- l136: the word "correctly" implies that there is a correct reference SIT, but here I think that you mean "similar" with and without anamorphosis. Otherwise, mention which reference is used.

Agreed with the comment, now it reads "the spatial structure is similar in the two cases, while the magnitude slightly differs "

- Figure 9: the colour shade for 1 and open water look the same to me (maybe because I am colour blind). Can you pick a red share instead?

Thank you for pointing out, the color palettes will be changed in all the figures to easy the Readers.

- Figures 8 and 9, the label CRYO2SMOS does not correspond to the name of the experiment in the text.

Thank you, corrected.

Typos:

- l1: Cryosphere

- l65: context

We corrected both, thank you.

- l79: coupling among or coupling between?

"Coupling between" is used

- C-GLORS is sometimes spelled CGLORS without the dash.

This is something we should have known, we correct to C-GLORS, thank you

- l114: Define the acronym CVT.

Thank you, corrected

- Use a capital letter for Gaussian since it comes from a person's name.

We now use the capital letter everywhere. Formally, being Gauss a person's name, the correct form is the possessive, i.e. Gauss'. The use of the adjective forms probably followed the same evolution discussed by Wright for Green function,

Wright, M. "Green function or green's function?", *Nature Phys* **2**, 646 (2006). https://doi.org/10.1038/nphys411

Reference:

Simon, E., & Bertino, L. (2012). Gaussian anamorphosis extension of the DEnKF for combined state parameter estimation: Application to a 1D ocean ecosystem model. Journal of Marine Systems, 89(1), 1–18. https://doi.org/doi:10.1016/j.jmarsys.2011.07.007
**Citation**: https://doi.org/10.5194/egusphere-2023-254-CC1